# Human Pegivirus-1 Detection and Genotyping in Brazilian Patients with Fulminant Hepatitis

**DOI:** 10.3390/pathogens12091122

**Published:** 2023-09-01

**Authors:** Anielly Sarana da Silva, Gabriel Montenegro de Campos, Marcia Guimarães Villanova, Rafael dos Santos Bezerra, Luciana Maria Mendes Santiago, Rodrigo Haddad, Dimas Tadeu Covas, Marta Giovanetti, Luiz Carlos Junior Alcantara, Maria Carolina Elias, Sandra Coccuzzo Sampaio, Simone Kashima, Svetoslav Nanev Slavov

**Affiliations:** 1Blood Center of Ribeirão Preto, Faculty of Medicine of Ribeirão Preto, University of São Paulo, Ribeirão Preto 14051-140, SP, Brazil; anielly.silva@unesp.br (A.S.d.S.); gabrielmdecampos@usp.br (G.M.d.C.); rafaelbezerra50@yahoo.com (R.d.S.B.); luciana.santiago@hemocentro.fmrp.usp.br (L.M.M.S.); dimas@fmrp.usp.br (D.T.C.); skashima@hemocentro.fmrp.usp.br (S.K.); 2Department of Gastroenterology, University Hospital, Faculty of Medicine of Ribeirão Preto, University of São Paulo, Ribeirão Preto 14048-900, SP, Brazil; villanova@hcrp.usp.br; 3Faculty of Ceilândia, University of Brasília, Brasília 72220-275, DF, Brazil; haddad@unb.br; 4Instituto Rene Rachou, Fundação Oswaldo Cruz, Belo Horizonte 30190-009, MG, Brazil; giovanetti.marta@gmail.com; 5Sciences and Technologies for Sustainable Development and One Health, Università Campus Bio-Medico di Roma, 00128 Rome, Italy; alcantaraluiz42@gmail.com; 6Center for Scientific Development, Butantan Institute, São Paulo 05503-900, SP, Brazil; carolina.eliassabbaga@butantan.gov.br (M.C.E.); sandra.coccuzzo@butantan.gov.br (S.C.S.)

**Keywords:** fulminant hepatitis, human pegvirus-1, HPgV-1, genotyping, qPCR

## Abstract

Fulminant hepatitis is a severe clinical disease characterized by a marked decline in liver function and encephalopathy. In a previous survey, using metagenomics in a group of 27 patients with this clinical condition, we observed an expressive quantity of reads of the Human pegivirus-1 (HPgV-1). Therefore, the objective of this study was to evaluate the frequency, molecular features, and HPgV-1 circulating genotypes in patients with fulminant hepatitis. After testing the collected plasma samples, we discovered twelve samples (44.4%) that were positive for HPgV-1 RNA (using both real-time and nested PCR). The positive samples presented a mean cycle threshold (Ct) of 28.5 (±7.3). Genotyping assignments revealed that all HPgV-1 positive samples belonged to the HPgV-1 genotype 2 (both subgenotypes 2A and 2B were identified). Although HPgV-1 is considered a commensal virus, little is known regarding its prevalence and genotypes in cases of fulminant hepatitis. More research is needed to understand whether HPgV-1 can be implicated in clinical disorders and infectious diseases.

## 1. Introduction

Fulminant hepatitis (FH) is a severe life-threatening disease characterized by rapid loss of hepatic function in people who have no history of previous hepatic disease. Clinical manifestations of FH include rapid hepatic injury, coagulation disturbances, hepatic encephalopathy, and there are rare cases of multiorgan failure [1].

The etiology of this condition is largely unknown, although the ingestion of drugs, mainly acetaminophen [2,3]; autoimmune or metabolic destruction of the hepatocytes [4]; and viral hepatotropic infections have been largely suggested as FH causes.

Viral agents play an important role in the etiology of FH and released data showed that viral etiology accounts for 12% of all FH cases in the United States [5]. One of the most important viral causes for FH remains the hepatitis A virus (HAV), but the hepatitis E virus (HEV) has also been implicated as a cause of FH, especially in pregnant women [5,6].

Human pegivirus-1 (HPgV-1) is considered a commensal virus that belongs to the *Flaviviridae* family [7]. Due to the fact that it was initially detected in samples obtained from patients with acute hepatitis, it was supposed that HPgV-1 is a causative agent of acute hepatic conditions [8]. However, more detailed studies demonstrated that it has no significant pathologic impact and is widely distributed throughout the world, including in healthy subjects who donate blood [9,10]. Additionally, it was proposed that HPgV-1 positivity in HIV-infected patients and in patients who have undergone hematopoietic transplantation might exert a positive impact on immune reconstitution [11,12].

During a previous metagenomic study that identified viral causes in cases of FH with no known etiology, we found the universal presence of HPgV-1 sequence reads. This finding raised questions related to the frequency and circulating genotypes among those patients. Therefore, a total of 27 plasma samples were tested for HPgV-1 RNA using RT-qPCR, sequencing, and phylogenetic analysis.

## 2. Materials and Methods

### 2.1. HPgV-1 RNA Detection Using Real-Time PCR

In this study, we tested for the presence of HPgV-1 RNA 27 plasma samples obtained from patients diagnosed with acute hepatitis (13 males and 14 females) who were diagnosed in the Ambulatory of Gastroenterology of the Clinical Hospital of Ribeirão Preto, Faculty of Medicine of Ribeirão Preto, University of São Paulo. The samples were obtained during the period June 2015–2017. The study was approved by the Institutional Ethics Committee of the University Hospital of Ribeirão Preto, University of São Paulo (process number CAAE 68164423.2.0000.5440). Each sample was submitted to manual viral nucleic acid extraction using 140 μL of plasma with the QIAamp Viral RNA MiniKit (QIAGEN, São Paulo, Brazil) following the manufacturer’s instructions. HPgV-1 real-time PCR detection was performed using primers and probes available in the literature [13,14]. The following sequences were applied: forward primer (5′-GGCGACCGGCCAAAA-3′), reverse primer (5′-CTTAAGACCCACCTATAGTGGCTACC-3′), and the hydrolytic probe 5′-FAM-TGACCGGGATTTACGACCTACCAACCCT-TAMRA-3′RNA. The amplification was performed using the GoTaq Probe 1-Step RT-qPCR System (Promega, Madison, WI, USA) with 400 nM of each primer and 200 nM of the probe in a 20 μL final reaction volume. The cycle protocol included 40 min at 40 °C for reverse transcription, 2 min at 95 °C for denaturation, and 40 cycles consisting of steps of 95 °C for 15 s and 60 °C for 1 min combining denaturation and annealing. The cycle threshold of the positive samples was automatically estimated.

### 2.2. HPgV-1 Genotyping of the 5′ Untranslated Region (5′-UTR)

The HPgV-1 genotype was determined in all positive samples using Sanger sequencing in the hypervariable 5′-untranslated region (5′-UTR), which is able to differentiate HPgV-1 genotypes from 1 to 7 [15]. The detection of HPgV-1 5′-UTR was performed using nested PCR with primers previously described in the literature [15]. Due to differences in the sensitivity of the real-time PCR and the nested PCR, we submitted all clinical samples to testing with the 5′-UTR nested PCR. Reverse transcription was carried out in a separate reaction using the High-Capacity cDNA Reverse Transcription Kit (ThermoFisher Scientific, Waltham, MA, USA) following the manufacturer’s instructions, applying the reverse primer of the first reaction. In brief, the primers used for the nested PCR detection were (i) first reaction: forward primer GUTRF-1 (5′-GGTTGGTAGGTCGTAAATCCCG-3´) and reverse primer GUTRR-1 (5′-GGTTGGTAGGTCGTAAATCCCG-3′) and (ii) second reaction: forward primer GUTRF-2 (5′-GTAGGTCGTAAATCCCGGTCA-3′) and reverse primer GUTRR-2 (5′-CGAAGGATTCTTGGGCTACC-3′) in a final volume of 50 μL. The reaction contained 1X Taq DNA polymerase PCR buffer, 1.5 mM of MgCl_2_, 400 nM of each primer, 200 μM of dNTP mix, and 1.25 U of Taq DNA polymerase (ThermoFisher Scientific). The amplification was performed in a SimpliAmp Thermal Cycler (ThermoFisher Scientific) using the following protocol: initial denaturation at 95 °C for 5 min, and then 40 and 35 cycles (first and second reaction, respectively) composed of 95 °C for 30 s (denaturation), 55 °C for 30 s (annealing), and extension at 72 °C for 1 min (extension), terminating with an elongation for 10 min. The amplicons were visualized in 2% agarose gel using a ChemiDocTM XRS Optical System (Bio-Rad, Hercules, CA, USA).

The sequencing of the 5′-UTR region was performed in an ABI 3500 XL DNA sequencer using the BigDye™ Terminator v.3.1 Cycle Sequencing Kit (ThermoFisher Scientific) with the following sequencing protocol: initial denaturation at 95 °C for 1 min and then 25 cycles of 96 °C for 10 s, 50 °C for 5 s, and 60 °C for 4 min.

### 2.3. Phylogenetic Analysis

The obtained sequence fragments were aligned with 87 genome sequences collected from the NCBI nucleotide database using the following keywords: “GB virus C complete genome” and “Human Pegivirus complete genome” (see Appendix A). Only sequences that were found to infect *Homo sapiens* were retrieved. The alignment was performed using MAFFT [16] and manually edited using Aliview [17]. The maximum-likelihood inference was performed using IQ-TREE software [18]; with bootstrap support of 1000 replicates, the Model Finder Plus (MFP) was used to find the most adequate nucleotide substitution model. Tree visualization was performed using Figtree [19].

## 3. Results

The samples received from patients with FH were initially investigated for the presence of HPgV-1 RNA using real-time PCR. We obtained positive results for four of them (*n* = 4/27 samples, 14.8%). The automatic amplification thresholds of the samples were 23.1, 23.6, 24.47, and 31.47 (mean 25.6 ± 3.9). The results obtained from the real-time amplification of HPgV-1 are shown in Figure 1A. We also submitted all collected samples to nested PCR testing with the objective to genotype the 5′-UTR portion of the viral genome. Positive results were obtained for 12 samples (44.4%, *n* = 12/27), which suggests a higher sensitivity of this reaction. Nested PCR amplification of the HPgV-1 378 bp amplicon is shown in Figure 1B. The differences in the HPgV-1 diagnosis between the real-time and nested PCR are shown in Table 1.

We also performed a phylogenetic analysis of all HPgV-1-generated sequences. The best model, which was chosen using the Bayesian information criterion (BIC), for phylogenetic reconstruction was TVMe+R4. Four of the twelve positive samples were classified as subgenotype 2A, while the remaining eight were classified as subgenotype 2B. All twelve samples belonged to genotype 2, which is the most prevalent in the world. All samples that were characterized as subgenotype 2B formed a separate cluster with sequences obtained from Brazil, the USA, and Egypt. The sequences that belonged to subgenotype 2A were also clustered together and were located in a cluster that included only genome sequences obtained from Brazil. Figure 2 depicts the HPgV-1 phylogenetic tree incorporating samples acquired from FH patients.

## 4. Discussion

This work, which looked at the frequency and genotyping of HPgV-1 in FH patients, was inspired by the previous metagenomic identification of reads belonging to this virus in clinical samples acquired from patients with FH. This naturally aroused concerns about the frequency of HPgV-1 in this severe clinical illness and the genotypes that circulate among the affected patients.

The estimated frequency of HPgV-1 RNA (real-time PCR and nested PCR) in the tested patients was 44.4% (n = 12/27). There is scarce information in the literature in regard to the prevalence of HPgV-1 in FH patients. In a study performed in India, the prevalence of HPgV-1 RNA among FH patients was 16%, which was lower compared with our results [20]. We believe that this difference might have been related to differences in the sensitivity of the applied HPgV-1 diagnostic tests or reflected regional variations in the HPgV-1 circulation rates. Since we applied a combination of diagnostic reactions, this strategy might have increased the detection sensitivity. Therefore, it is highly possible that we detected very low quantities of circulating HPgV-1 RNA, thus increasing the observed frequency. We are also unaware whether such a finding represents an infectious virus since to confirm this would need HPgV-1 cultivation, which is a cumbersome process [21] and was not the objective of this investigation.

A limitation of our study was that we did not include an accompanying negative control. However, a study performed by our group revealed an HPgV-1 RNA prevalence of 12.5% in blood donors from the same geographic region and using the same diagnostic reaction [22]. Additionally, the global HPgV-1 prevalence among healthy blood donors was reported to be 3.1% according to a recently published meta-analysis [23]. These results show that HPgV-1 could demonstrate higher circulation among FH patients but further studies are necessary to reveal whether such an association exists.

The clinical significance of HPgV-1 as a cause of hepatic dysfunction is still unknown and we believe that in positive cases, HPgV-1 is an accidentally discovered bystander. Initially, it was believed that this virus, which has a genome similar to HCV, could be the cause of acute viral non-A, non-B hepatitis [23,24,25], but it was soon discovered that it is a commensal virus, like the anelloviruses, and thus, an essential part of the human blood virome [25]. Nonetheless, no studies have been conducted to investigate whether HPgV-1 can have a therapeutic impact, particularly through immune response modulation. Slower progression in HIV infection was previously associated with HPgV-1 presence, highlighting a possible beneficial effect [26,27]. A possible HPgV-1 beneficial impact was also reported in allogenic hematopoietic transplant recipients [11]. More detailed studies seem to be necessary to consolidate the clinical implications and HPgV-1 mechanisms of infection, especially in cases of co-infections with other clinically important viruses, as in the example of HIV [28].

We performed phylogenetic analysis on all positive samples in order to characterize the circulating HPgV-1 genotypes among FH patients. All our samples belonged to the HPgV-1 genotype 2: 67% belonging to subgenotype 2B and 33% belonging to subgenotype 2A. Genotype 2 is a prevalent genotype of HPgV-1, being the most disseminated in Europe and the USA [29,30,31]. It is particularly detected among individuals who inject drugs [10,32,33]. Studies performed in Brazil show that in this Latin American country, genotype 2 is dominant [9,31,34]; this includes the state of São Paulo [34]. This might be the reason for the extensive identification of HPgV-1 genotype 2 among the tested patients. An interesting observation in the phylogenetic tree was that the strains obtained from FH patients clustered together with the reference Brazilian sequences. Such a clustering might be due to a regional HPgV-1 circulation since RNA viruses present local genetic diversification related to a genotype-specific geographic distribution [31,35,36]. Therefore, the genotypic distribution observed in our study might have been a reflection of the circulation of the most pronounced genotype in the state and Brazil, which is important for studies connected to viral molecular epidemiology.

## 5. Conclusions

We observed a high frequency of HPgV-1 RNA among FH patients; however, we used a combination of strategies that could have detected low circulating viral loads. The most common genotype discovered in the analyzed samples was HPgV-1 genotype 2 (with a higher prevalence of subgenotype 2B), which is the most common genotype in Brazil. Several aspects of HPgV-1 infection and the significance of the virus to human pathophysiology remain unknown. More research is needed to determine the influence of this virus, if such exists, on human acute diseases and immune response modulation.

## Figures and Tables

**Figure 1 pathogens-12-01122-f001:**
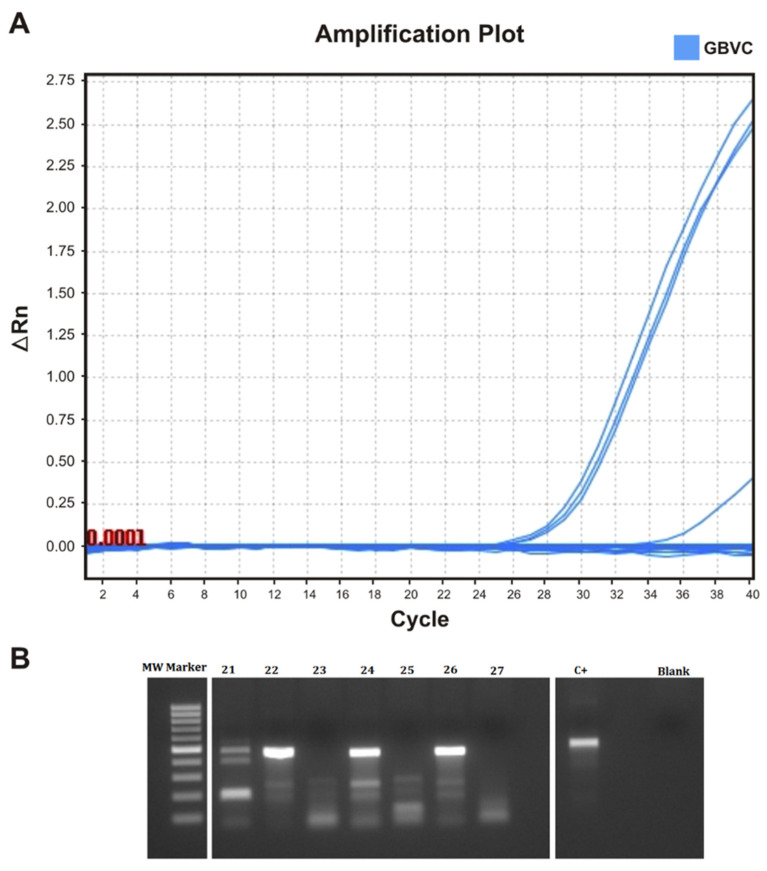
Detection of HPgV-1 using real-time PCR and nested PCR. (**A**) Detection curves of HPgV-1 RNA-positive individual samples obtained using RT-qPCR amplification. The cycle threshold was calculated automatically. (**B**) Agarose gel of the electrophoresis of some of the positive HPgV-1 amplicons of 378 bp (samples 22, 23, and 26). C+ positive control; MW marker: molecular weight marker.

**Figure 2 pathogens-12-01122-f002:**
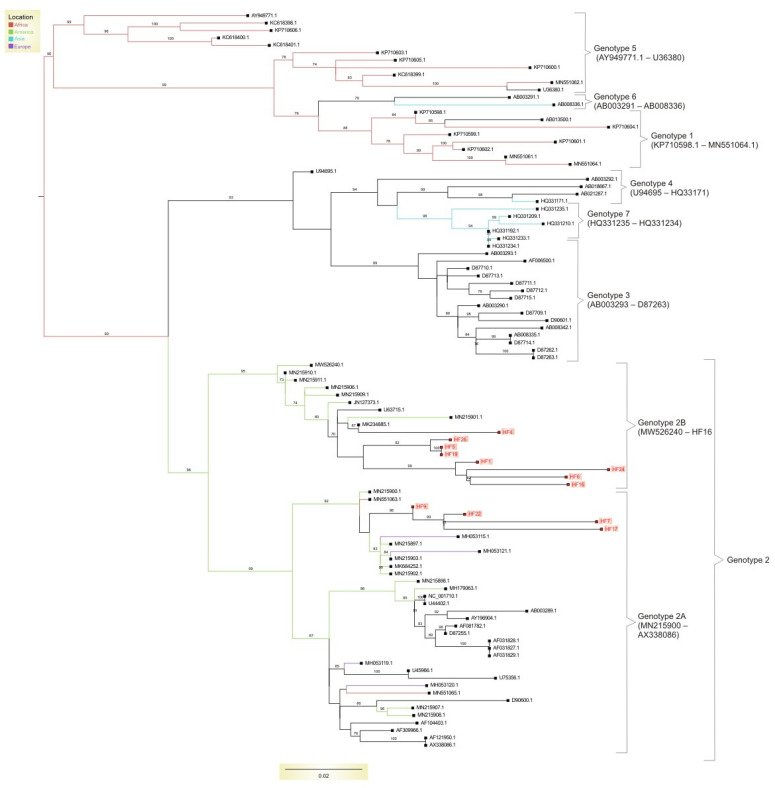
Maximum likelihood tree containing worldwide Human pegivirus-1 genotypes and the strains obtained from patients with fulminant hepatitis (FH). All samples were classified as belonging to genotype 2, which is the most widely spread HPgV-1 genotype. However, within genotype 2, most of our samples belonged to subgenotype 2B. In all cases, samples from the FH patients were clustered in different clusters, with the reference samples being primarily from Brazil; this was due to the fact that genotype 2 is a predominant genotype in Brazil. The obtained sequences were deposited in GenBank under the following accession numbers: OR227612–OR227623.

**Table 1 pathogens-12-01122-t001:** Nested and real-time PCR results for the positive HPgV-1 RNA samples.

Sample ID	Amplification Results		
	Nested-PCR	RT-PCR	Cycle Threshold
HF01	+	-	NA *
HF04	+	+	23.6
HF05	+	+	23.1
HF06	+	-	NA *
HF07	+	-	NA *
HF09	+	+	24.3
HF16	+	-	NA *
HF17	+	-	NA *
HF19	+	+	31.5
HF22	+	-	NA *
HF24	+	-	NA *
HF26	+	-	NA *

NA: not applicable; *: samples were tested using nested-PCR and were not amplified using real-time PCR.

## Data Availability

The obtained 5′-UTR HPgV-1 sequences were deposited under the following accession numbers: OR227612–OR227623.

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
