# Peer review of "Human Pegivirus-1 Detection and Genotyping in Brazilian Patients with Fulminant Hepatitis"

_pathogens, 2023, doi:10.3390/pathogens12091122_

Round 1
Reviewer 1 Report
This study by da Silva et al. reports the detection of HPgV-1 in 12 out of 27 Brazilian fulminant hepatitis patients. Overall, I found that the manuscript is poorly written, and is difficult to follow in many places. The results are minor, and are poorly presented, and discussed.
Comments
One of the major issues with the manuscript is that the study design is described very confusingly. For example, while the method section states that the virus was detected by using RT-PCR (Line 61-80), and was sequenced by using Sanger sequencing (Line 81 - 107), the authors instead describe in the title and the results section that the virus was identified by using next-generation metagenomic sequencing (Line 118 - 121). This is confusing, if not entirely misleading, as the main methodology used to derive the results (i.e. virus detection and genotyping) had very little to do with metagenomic sequencing, but PCR and sanger sequencing. It also appears that the samples were pooled together for the analysis for some reasons (Line 120), but the rationale behind the study design, and the details of the sample pooling, are not at all provided in the text.
The results are also poorly presented. For instance, the Ct values are reported only for some positive samples (Line 127), making the reported mean Ct value (28.5 ± 7.3, Line 27) virtually untraceable. Furthermore, while Line 125 – 128 implies that Figure 1 shows the results from the RT-PCR experiments for the first four RT-PCR positive samples, Lines 128 – 132 instead implies that Figure 1 shows the results from the secondary nested-PCR analysis of the RT-PCR negative samples in the “last pool”, which is, again, very confusing. Figure 1B is labelled so poorly that it is uninterpretable. The phylogeny (Figure 2) is also poorly illustrated, with illegible branch support values, a missing scale bar, and no information on the model used to construct the tree, or the method used to root the tree.
Lastly, citations are missing in several places (e.g. Line 44, 84, 181, 192, and 210), and the discussion is inappropriate in some instances. For example, the authors mention that the prevalence of HPgV-1 observed in their study is greater than that reported by a study from India (Line 166-167), but no statistical analysis was used to support their claim. The authors mention that their Brazilian HPgV-1 sequences clustered together with other Brazilian sequences on the tree, suggesting regional HPgV-1 circulation (which is reasonable), but they then reference a paper on long-term co-evolution between ancient human migrations and HPgV-1 origin by Pavesi (ref. 31), which is hardly relevant to their findings.
The manuscript is difficult to follow in many places due to occasional grammatical, spelling, and punctuation errors. Words are often chosen poorly sometimes.
Author Response
REPLY TO THE REVIEWERS COMMENTS/CRITICISMS
REVIEWER #1
This study by da Silva et al. reports the detection of HPgV-1 in 12 out of 27 Brazilian fulminant hepatitis patients. Overall, I found that the manuscript is poorly written, and is difficult to follow in many places. The results are minor, and are poorly presented, and discussed.
We are very sorry for the opinion of the reviewer. In the letter below and considering all the obtained comments, we tried to substantially improve the presentation of the obtained results, the discussion and the manuscript as a whole.
Comments
One of the major issues with the manuscript is that the study design is described very confusingly. For example, while the method section states that the virus was detected by using RT-PCR (Line 61-80), and was sequenced by using Sanger sequencing (Line 81 - 107), the authors instead describe in the title and the results section that the virus was identified by using next-generation metagenomic sequencing (Line 118 - 121). This is confusing, if not entirely misleading, as the main methodology used to derive the results (i.e. virus detection and genotyping) had very little to do with metagenomic sequencing, but PCR and sanger sequencing. It also appears that the samples were pooled together for the analysis for some reasons (Line 120), but the rationale behind the study design, and the details of the sample pooling, are not at all provided in the text.
We are grateful for the valuable comment of Reviewer#1.
We apologize for the not well-presented study design in the initial version of the manuscript. In fact, in a previous study, we performed a metagenomic analysis on the collected samples from patients with acute hepatitis and without etiological diagnosis. To perform the metagenomic analysis, we used a standard pooling procedure in order to reduce the cost of the sequencing. After pooling, we tested a total number of 4 pools containing 27 plasma samples obtained from patients with fulminant hepatitis. The samples were randomly distributed across the pools like the following: one pool contained 6 plasma samples and 3 pools were assembled from 7 plasma samples. The total volume of the pool was extracted using the High Pure Large Volume Nucleic Acids kit (Roche) and this was followed by reverse transcription, isothermal amplification of the cDNA, library preparation and sequencing. The initial bioinformatic analysis of the obtained sequencing data was performed using a specific pipeline focused on taxonomic classification of the obtained viral abundance. The analysis revealed that all of the pools contain relatively large sequence reads belonging to Human pegivirus-1 (HPgV-1). This focused our attention on the prevalence of this virus and its molecular characteristics in patients with fulminant hepatitis.
Therefore, using as a base the obtained metagenomic results, we decided to investigate the prevalence and molecular characteristics of the HPgV-1 infection (genotype, cycle threshold of amplification) among patients with fulminant hepatitis. We tested all individual samples composing the pools initially by real-time PCR using primers and probe that were already available in the literature (Souza, I.E. et al., 2006; Kriesel, J,D. et al.,2012). The initial testing of these samples rendered 4 positive results from 3 metagenomic pools. What puzzled us was that the individual samples that composed the fourth pool showed negative results for HPgV-1 RNA when tested by real-time PCR although it showed significant read number. That is why we decided to test all samples obtained from patients with fulminant hepatitis by nested-PCR using primers available in the literature (Miao, Z. et al.,2017). Nested-PCR is more sensitive and can detect very low quantities of circulating nucleic acids. We obtained 3 positive results for this pool. All of the positive results obtained for real-time PCR were confirmed and we detected additionally 5 samples that were positive for HPgV-1. To more clearly present this, we added the following table in the manuscript that describes the samples and the obtained results by real-time and nested-PCR.
|
Table 01. Nested and real-time PCR results for the positive for HPgV-1 RNA samples |
|||
|
Sample ID |
Amplification results |
||
|
Nested-PCR |
RT-PCR |
Cycle threshold |
|
|
HF01 |
+ |
_ |
NA* |
|
HF04 |
+ |
+ |
23.6 |
|
HF05 |
+ |
+ |
23.1 |
|
HF06 |
+ |
_ |
NA* |
|
HF07 |
+ |
_ |
NA* |
|
HF09 |
+ |
+ |
24.3 |
|
HF16 |
+ |
_ |
NA* |
|
HF17 |
+ |
_ |
NA* |
|
HF19 |
+ |
+ |
31.5 |
|
HF22 |
+ |
_ |
NA* |
|
HF24 |
+ |
_ |
NA* |
|
HF26 |
+ |
_ |
NA* |
|
*NA: not applicable for the samples that were tested by nested-PCR and did not amplify by real-time PCR |
|||
Using Sanger sequencing we determined the HPgV-1 genotype of all positive samples. We totally agree with the reviewer that the title of the manuscript did not correctly reflect what was performed in this work. We therefore, performed the following changes in the title and the manuscript itself:
The title was changed with the removal of the word metagenomics and now we believe that it more correctly reflects what was performed in this investigation:
Line 2-3, Page 1: The previous title of the manuscript ” Human pegivirus-1 detection and genotyping in Brazilian fulminant hepatitis patients using metagenomics” was changed to “Human pegivirus-1 detection and genotyping in Brazilian patients with fulminant hepatitis”
We also performed the following modifications in this manuscript:
We explained in better that the metagenomics was used in a previous study that was used as a base for this study:
Lines 54-58, Page 2 “... During a previous metagenomic study identifying viral causes in cases of FH with no known etiology, we found universal presence of HPgV-1 sequence reads. This finding raised questions related to the prevalence and circulating genotypes among those pa-tients. Therefore, a total of 27 plasma samples were tested for HPgV-1 RNA using RT-qPCR, sequencing and phylogenetic analysis.…”
Lines 67-69, Page 2: We deleted from the materials and methods the pool assembly that was not an objective of this study:
“... Each individual sample was submitted to manual viral RNA extraction using 140 μL of plasma by the QIAamp Viral RNA MiniKit (QIAGEN, São Paulo, Brazil), following the manufacturer’s instructions. …”
Lines 118-119, Page 2: we deleted from the results section all phrases where appears “pools” in order not to cause misunderstanding with the metagenomic studies.
“... In all the four pools tested we found positivity to HPgV-1 reaching mean number of 3405 reads (range 1198 to 8725). …”
Lines 119-121, Page 3: We removed from the text “viral load” as we did not perform quantification and only evaluated the cycle threshold of this infection
“... This led to inquiries on the molecular detection of HPgV-1 and its genotypes in such patients, among other molecular aspects associated to the HPgV-1 infection. …”
The results are also poorly presented. For instance, the Ct values are reported only for some positive samples (Line 127), making the reported mean Ct value (28.5 ± 7.3, Line 27) virtually untraceable. Furthermore, while Line 125 – 128 implies that Figure 1 shows the results from the RT-PCR experiments for the first four RT-PCR positive samples, Lines 128 – 132 instead implies that Figure 1 shows the results from the secondary nested-PCR analysis of the RT-PCR negative samples in the “last pool”, which is, again, very confusing. Figure 1B is labelled so poorly that it is uninterpretable. The phylogeny (Figure 2) is also poorly illustrated, with illegible branch support values, a missing scale bar, and no information on the model used to construct the tree, or the method used to root the tree.
We are sorry that the reviewer did not agree with the presentation of the results. We gave our best to better present the obtained results.
In response to the reviewer regarding the discordance of the obtained results, as explained above, we observed that by real-time PCR testing positive results were obtained for the first three sample pools and no positive result was obtained for the fourth pool when the individual samples were tested for HPgV-1. That is why we applied additional nested-PCR testing that shows higher sensitivity compared to real-time PCR to all samples. Nested-PCR was also used for the genotyping of HPgV-1. We performed modifications in the text excluding as pointed by the reviewer “the fourth pool”. We greatly shortened this part of the “Results section”:
Lines 122-129, Page 3 “… As a result, all samples received from patients with FH were initially investigated for the presence of HPgV-1 RNA using real-time PCR. We obtained positive results for four of them (n=4/27 samples, 14.8%). The automatic amplification thresholds of the samples were 23.1, 23.6, 24.47 and 31.47. The results obtained from the real-time am-plification of HPgV-1 are shown in Figure 1A. We also submitted all collected samples to nested-PCR testing aiming genotyping of 5´-UTR portion of the viral genome. Posi-tive results were obtained for 12 samples (44.4%, n=12/27) that suggests higher sensi-tivity of this reaction. Nested-PCR amplification of the HPgV-1 378 bp amplicon is shown in Figure 1B. …”
We also performed modifications regarding the presentation of the phylogenetic results. Applying a ModelFider Plus (MFP) as implied by IQ-Tree program, the best substitution-fit model chosen for our analysis was the TVMe+R4 according the Bayesian Information Criterion (BIC). According to IQ-Tree documentation, TVMe stands for Transversion Model equal and it is a DNA model, therefore AG = CT and equal base frequency. +R is a FreeRate model that generalizes the +G model (discrete Gamma model) by relaxing the assumption of Gamma-distributed rates and is used to rate heterogeneity across sites and 4, default value, is the number of categories.
To root the tree, we used the FigTree program, where we rooted it by its midpoint and by decreasing the nodes. We also removed the values that were below 70% from the tree branches. We added the following information in the text:
Lines 126-128, Page 3: We removed from the text the phrase that caused misunderstanding.
“...Considering the fact that real-time PCR did not detect any HPgV-1 RNA in the samples from the last pool and it was positive for HPgV-1 by the metagenomics, we submitted the individual samples composing this pool to the 5´-UTR nested-PCR. …”
Lines 127-131, Page 3: We removed all portion from the text that explains the additional testing by nested-PCR and we presented this data as a part of the genotyping process:
“... Considering the fact that real-time PCR did not detect any HPgV-1 RNA in the samples from the last pool and it was positive for HPgV-1 by the metagenomics, we submitted the individual samples composing this pool to the 5´-UTR nested-PCR. The application of nested-PCR was able to detect HPgV-1 in three samples. …”
We deleted all the part that described the testing of the negative by real-time PCR samples
Lines 128-133, Page 3: “...This result suggested higher sensitivity of the applied nested-PCR that more samples might be positive for HPgV-1 RNA. Due to this, we tested all negative samples with real-time PCR. Figure 1B depicts positive samples for HPgV-1 detected by nested-PCR reaching an overall frequency of HPgV-1 of 44.4% (n=12/27) among 27 samples obtained from patients with FH. …”
We inserted in the “Results” section the model that was used for phylogenetic tree reconstruction
Lines 131-133, Page 4: “... The best model that was chosen by the Bayesian Information Criteria (BIC) for phylo-genetic reconstruction was the TVMe+R4. …”
The newly modified phylogenetic tree is shown in Figure 2.
Lastly, citations are missing in several places (e.g. Line 44, 84, 181, 192, and 210), and the discussion is inappropriate in some instances. For example, the authors mention that the prevalence of HPgV-1 observed in their study is greater than that reported by a study from India (Line 166-167), but no statistical analysis was used to support their claim. The authors mention that their Brazilian HPgV-1 sequences clustered together with other Brazilian sequences on the tree, suggesting regional HPgV-1 circulation (which is reasonable), but they then reference a paper on long-term co-evolution between ancient human migrations and HPgV-1 origin by Pavesi (ref. 31), which is hardly relevant to their findings.
We are grateful for the comments of Reviewer#1
We examined the text and especially the lines that the reviewer suggested that there are missing references. We included the following citations:
Line 43: We added the following references that describe the causes for fulminant hepatitis worldwide:
Schiødt, F.V.; Davern, T.J.; Shakil, A.O.; McGuire, B.; Samuel, G; Lee, W.M. Viral hepatitis-related acute liver failure. Am J 247 Gastroenterol 2003, 98, 448-53. https://doi.org/10.1111/j.1572-0241.2003.t01-1-07223.x. (Reference N°5)
Line 181: We included the following reference:
Yu, Y.; Wan, Z.; Wang, J.H.; Yang, X.; Zhang, C. Review of human pegivirus: Prevalence, transmission, pathogenesis, and clinical implication. Virulence 2022, 13, 324-341 (reference N° 26);. https://doi.org/10.1080/21505594.2022.2029328.
Line 196: we distributed the citations and added the following citations:
- Zimmerman J, Blackard JT. Human pegivirus type 1 infection in Asia-A review of the literature. Rev Med Virol. 2022 Jan;32(1):e2257. doi: 10.1002/rmv.2257. Epub 2021 May 26. PMID: 34038600 (reference N° 30).
- Niama RF, Mayengue PI, Nsoukoula BCM, Koukouikila-Kossounda F, Badzi CN, Mandiangou AF, Louzolo I, Fila-Fila GPU, Dossou-Yovo LR, Angounda MB. Genetic variability of human pegivirus type 1 (HPgV-1) among Congolese co-infected with hepatitis C virus in Brazzaville, Congo. IJID Reg. 2023 Mar 21;7:191-192. doi: 10.1016/j.ijregi.2023.03.005. PMID: 37123381; PMCID: PMC10131043 (reference N°31).
- Slavov, S.N.; Maraninchi Silveira, R.; Hespanhol, M.R.; Sauvage, V.; Rodrigues, E.S.; Fontanari Krause, L.; Bittencourt, H.T.; Caro, V.; Laperche, S; Covas, D.T.; Kashima, S. Human pegivirus-1 (HPgV-1) RNA prevalence and genotypes in volunteer blood donors from the Brazilian Amazon. Transfus Clin Biol 2019, 26, 234-239. https://doi.org/10.1016/j.tracli.2019.06.005 (reference N° 32).
Line 210: We did not added references in the conclusion section
In our discussion section we compared the obtained results in our study with the only study available describing the prevalence of HPgV-1 among patients with fulminant hepatitis in India. As this comparison was performed on the basis of our obtained frequency, we only speculate in the discussion section the reasons that can lead to lower or higher prevalence of this virus in the respective patients. We believe that at least in this case, there is no need to perform statistical confirmation. Additionally, we included a statement about the control group. We performed the following modifications in the text:
Lines 172-178, Page 5: “… Our study was a descriptive one and we did not include in parallel a negative control. However, a study performed by our group revealed a HPgV-1 RNA prevalence of 12.5% in blood donors from the same geographic region and using the same diag-nostic reaction [22]. Additionally, the global HPgV-1 prevalence among healthy blood donors was reported to be 3.1% according to a recently published meta-analysis [23]. These results show that HPgV-1 might demonstrate higher circulation among FH pa-tients but further studies are necessary to reveal if such association exits. …”
In the performed HPgV-1 phylogenetic analysis, the obtained sequences were clustered with reference genomes predominantly from Brazil. We would like to stress that the phylogenetic reconstruction was only based on selection of complete genomes that were consequently aligned to our fragment and by that type of dataset preparation we show the main evolutionary lines of the obtained sequences (partial HPgV-1 were not included in our dataset).
The observed clustering of the sequenced samples with other Brazilian sequences suggests a regional circulation of the detected strains and regional evolution that is typical for HPgV-1. We supposed that the regional circulation of HPgV-1 might be also a result of co-evolution of HPgV-1 with its host in this Latin America region, leading to regional clustering of the obtained HPgV-1 sequences. This is the reason why we cited the article of Pavesi et al. 2001. Nevertheless, we also added citations that describe the circulation of HPgV-1 genotype 2 in Brazil that reflect the regional circulation of HPgV-1. We excluded the reference Pavesi et al., 2001. The following modifications were performed in text:
Lines 197-206, Page 6: We presented in a better way the genotypic circulation of HPgV-1 genotype 2 among patients with fulminant hepatitis
Studies performed in Brazil, show also that in the Latin American country genotype 2 is a dominant one [9,32,35] including the state of São Paulo [35]. This might be the reason, for the extensive identification of HPgV-1 genotype 2 among the tested pa-tients. An interesting observation in the phylogenetic tree was that the strains obtained from FH patients clustered together with reference Brazilian sequences. Such a clustering might be due to a regional HPgV-1 circulation, since RNA viruses present local genetic diversification related genotype-specific geographic distribution [32,36]. Therefore, the genotypic distribution observed in our study might be a reflection of the circulation of the most pronounced genotype in state and in Brazil that is important for studies connected to viral molecular epidemiology.
We deleted the following phrase: “... As an ancient virus, several hypotheses can be raised about its evolutionary history, e.g., co-evolution with the host, but there is yet much information to be studied. …”
Reviewer 2 Report
This is an interesting paper describing the prevalence of HPgV-1 RNA in sera from humans experiencing fulminant hepatitis. The authors describe the RNA copy numbers in 27 patients having fulminant hepatitis and perform a phylogenetic analysis of the RNA found in these individuals. The manuscript concludes that there is a high number of reads of HPgV-1 in patients with fulminant hepatitis, though the authors do not state against whom they are comparing that this number of reads is high, and suggests more research on HPgV-1 is needed.
Major comments:
-
There is no control group for comparison of the number of reads of HPgV-1 in people not experiencing fulminant hepatitis. At the very least this comparison should be done for healthy controls, and it would be best to have an additional comparison group composed of individuals having non-fulminant hepatitis or some other related condition. Without at least a healthy control group, the data presented here are impossible to interpret. As an example: in lines 160-161, the authors state this work was inspired by the “large number of reads of this virus in clinical samples acquired from patients with FH”---but how “large” is being determined here is unknown, since no comparison or control group is described.
-
The type of phylogenetic analysis used, and the program used to conduct the phylogenetic analysis, need to be described in the Methods.
-
There is no discussion around why the authors only describe RNA copy numbers and did not culture the virus (the virus is not known to be culturable, but this needs to be explicitly stated somewhere by the authors).
-
Lines 183-184, the statement “Many studies have proven the positive effect of HPgV-1 in HIV infection due to its slower progression [24,25].” is an incorrect interpretation of these data. Many studies have shown an association of HPgV-1 infection and slower progression of HIV, but no study has investigated whether there is any causative relationship between HPgV-1 and slower HIV progression. There is no evidence that HPgV-1 has any role in slowing HIV infection: this is only an association, and may well be due to other confounding factors.
-
Lines 185-188 make a recommendation that does not seem to follow from any data, evidence, or review presented in the paper: there is no evidence of a positive effect of HPgV-1, but rather only an association with slower progression of HIV. The statement “In our opinion, and especially regarding the obtained data, we recommend additional extensive studies of the potential clinical implications of HPgV-1, particularly in cases of co-infections with other clinically relevant viruses such as HIV [26].” should be significantly tempered to reflect this lack of data or evidence, or this statement should be removed entirely.
Author Response
REPLY TO THE REVIEWERS COMMENTS/CRITICISMS
REVIEWER #2
This is an interesting paper describing the prevalence of HPgV-1 RNA in sera from humans experiencing fulminant hepatitis. The authors describe the RNA copy numbers in 27 patients having fulminant hepatitis and perform a phylogenetic analysis of the RNA found in these individuals. The manuscript concludes that there is a high number of reads of HPgV-1 in patients with fulminant hepatitis, though the authors do not state against whom they are comparing that this number of reads is high, and suggests more research on HPgV-1 is needed.
In first place we would like to thank to reviewer #2 for the positive feedback considering our study. Thank you very much!
We would also like to thank to reviewer#2 for the valuable comments. Regarding the HPgV-1 RNA copy numbers, we actually refer to the number of generated reads obtained from this virus by next-generation sequencing in the previous metagenomics analysis. These results are not an objective of this study as they only serve for the assumption that HPgV-1 might be found in patients with fulminant hepatitis.
We apologize for the misunderstanding that was caused by the used terminology and we changed all terms in the text that can cause misunderstanding. The same we did with the term “high number of reads”, since there is no direct comparison between the groups. The following modifications were performed in the text:
Lines 54-58, Page 2: We showed that the obtained HPgV-1 reads were from a previous metagenomic survey “... During a previous metagenomic study identifying viral causes in cases of FH with no known etiology, we found universal presence of HPgV-1 sequence reads. …”
We totally modified our results section, giving emphasis on the results that were obtained from this study:
Lines 117-121, Page 3: We explained better why we tested for HPgV-1 the samples originating from patients with fulminant hepatitis “... The performed study was based on previous metagenomic identification of HPgV-1 reads by next-generation sequencing in pooled plasma samples belonging to patients with FH (range of reads between 1198 to 8725). This led to inquiries on the molecular detection of HPgV-1 and its genotypes in such patients, among other molec-ular aspects associated to the HPgV-1 infection. ….”
Major comments:
There is no control group for comparison of the number of reads of HPgV-1 in people not experiencing fulminant hepatitis. At the very least this comparison should be done for healthy controls, and it would be best to have an additional comparison group composed of individuals having non-fulminant hepatitis or some other related condition. Without at least a healthy control group, the data presented here are impossible to interpret. As an example: in lines 160-161, the authors state this work was inspired by the “large number of reads of this virus in clinical samples acquired from patients with FH”---but how “large” is being determined here is unknown, since no comparison or control group is described.
Still on this matter, the performed study actually did not imply the comparison with a healthy group of patients once the metagenomic analysis was performed only on patients with fulminant hepatitis. However, to respond the comment of the reviewer and to perform comparison with healthy controls we cited a study published by our group and which used a group of healthy blood donors as control in the same region (Valença I et al., 2021). In this study 16 healthy blood donors were tested for the presence of HPgV-1 RNA and genetic material of this virus was found in 2 of them (12.5%) of the participants. This information will be added in the discussion, lines 166-170, alongside with other references for the prevalence of HPgV-1 in healthy populations in Brazil.
Lines 172-178, Page 5. The following modifications in the discussion section were performed:
“... Our study was a descriptive one and we did not include in parallel a negative control. However, a study performed by our group revealed a HPgV-1 RNA prevalence of 12.5% in blood donors from the same geographic region and using the same diag-nostic reaction [22]. Additionally, the global HPgV-1 prevalence among healthy blood donors was reported to be 3.1% according to a recently published meta-analysis [23]. These results show that HPgV-1 might demonstrate higher circulation among FH pa-tients but further studies are necessary to reveal if such association exits. …”
The type of phylogenetic analysis used, and the program used to conduct the phylogenetic analysis, need to be described in the Methods.
The sequenced fragments of the twelve 5´ UTR obtained sequences were aligned along with 87 complete genomes retrieved from NCBI nucleotide database using the following keywords: “GB virus C complete genome” and “Human Pegivirus complete genome”; we filtered for the sequences for host Homo sapiens only. We also used, as reference genotypes to classify the obtained sequences, the Accession Number provided by Slavov et al., in the paper doi: 10.1016/j.transci.2019.01.002. We did not used a unique program to perform phylogenetic analysis but rather a command line and a pipeline. The alignment was performed using MAFFT (Katoh; Standley, 2013) and manually edited using Aliview (Larsson, 2014). To perform the phylogenetic analysis presented as a cladogram, our group utilized the IQ-Tree software (Nguyen et al., 2015) in order to apply maximum-likelihood inference with bootstrap support of 1,000 replicates. Visualization and editing of the phylogenetic tree was performed using Figtree v1.4.4 (Rambaut, 2010). We added this information in the “Materials and Methods” section:
Lines 110-115, Page 3. “...Only sequences that were found to infect Homo sapiens were retrieved. The alignment was performed using MAFFT [16] and manually edited using Aliview [17]. The maxi-mum-likelihood inference was performed using the IQ-TREE software [18], with boot-strap support of 1,000 replicates, the Model Finder Plus (MFP) was used to find the most adequate nucleotide substitution model. Tree visualization was performed using Figtree [19]. …”
There is no discussion around why the authors only describe RNA copy numbers and did not culture the virus (the virus is not known to be culturable, but this needs to be explicitly stated somewhere by the authors).
We are grateful for the valuable comment of reviewer #1.
The objective of this article was a descriptive study that investigated the prevalence and molecular characteristics of HPgV-1 among patients with fulminant hepatitis. We only mentioned the number of HPgV-1 reads as a result obtained by a previous metagenomic analysis that was the base of our study. HPgV-1 is not a cultivable virus and therefore changes were performed in the discussion section, addressing this information in 168-170.
“… We are also unaware if such a finding represents an infectious virus, once to confirm this, a HPgV-1 cultivation is warranted but this is a cumbersome process [21] and was not an objective of this investigation. …”
As previously stated, the used terms generated misunderstanding around the interpretation of HPgV-1 reads. Therefore, we excluded from the manuscript text these statements and performed general modification of the terms. We performed the following modifications in the text:
Lines 165-167, Page 5: “...Once we applied a combination of diagnostic reactions, such a strategy might have in-creased the detection sensitivity. Therefore, it is highly possible that we have detected very low quantities of circulating HPgV-1 RNA, increasing thus the observed preva-lence. …”
Lines 183-184, the statement “Many studies have proven the positive effect of HPgV-1 in HIV infection due to its slower progression [24,25].” is an incorrect interpretation of these data. Many studies have shown an association of HPgV-1 infection and slower progression of HIV, but no study has investigated whether there is any causative relationship between HPgV-1 and slower HIV progression. There is no evidence that HPgV-1 has any role in slowing HIV infection: this is only an association, and may well be due to other confounding factors.
We are grateful for the valuable comment of reviewer #2.
We completely agree with the suggestion of the reviewer and in our revised version of this manuscript and especially our interpretation of the effect of HPgV-1 on the HIV progression. In accordance with the reviewer suggestion, we appropriately modified the sentences in lines 184-187 to fully comprehend the existence of associations between HPgV-1 and slower HIV infection progression, not causative effects.
Lines 184-187, Page 6 “... Nonetheless, no studies have been conducted to investigate whether HPgV-1 can have a therapeutic impact, particularly through immune response modulation. Slower pro-gression in HIV infection has been previously associated with HPgV-1 presence, high-lighting a possible beneficial effect [27,28]. …”
Lines 185-188 make a recommendation that does not seem to follow from any data, evidence, or review presented in the paper: there is no evidence of a positive effect of HPgV-1, but rather only an association with slower progression of HIV. The statement “In our opinion, and especially regarding the obtained data, we recommend additional extensive studies of the potential clinical implications of HPgV-1, particularly in cases of co-infections with other clinically relevant viruses such as HIV [26].” should be significantly tempered to reflect this lack of data or evidence, or this statement should be removed entirely.
The sentence in lines 185-188 has also been replaced since our conclusions do not provide enough information on this matter, as indicated by reviewer #2.
Lines 188-191 Page 6: “... More detailed studies seem to be necessary to consolidate the clinical implications and HPgV-1 mechanisms of infection, especially in cases of co-infections with other clini-cally important viruses as the example of HIV [29] …”
References for this reply:
- Valença, I.N.; Rós, F.A.; Zucherato, V.S.; Silva-Pinto, A.C.; Covas, D.T.; Kashima, S.; Slavov, S.N. Comparative metavirome analysis in polytransfused patients. Braz J Med Biol Res 2021, 54. https://doi.org/10.1590/1414-431X2021e11610.
- Rambaut, A. FigTree v1.3.1. Institute of Evolutionary Biology, University of Edinburgh, Edinburgh. Available online: http://tree.bio.ed.ac.uk/software/figtree (accessed on 01 04 2023).
Reviewer 3 Report
In this manuscript, authors present a study aimed to detected and characterize HPgV-1 form 27 patients diagnosed with severe hepatic failure. This work continues a supposed previous work in which, by a metagenomic approach, several reads belonging to HPgV-1 were detected.
HPgV-1 has been studied as a likely commensal virus, not specifically associated to a particular medical condition or pathological process, since it has been also identified in healthy individuals.
Further research is needed to clarify the potential (pathogenic and/or benefic) role of this agent in certain conditions. Several reports focused on HPgV-1 detection and characterization in several populations and health conditions has been published from Brazil. Even though this work might be of interest it needs to be substantially improved before considering it for publication.
Major comments
This work was not performed with a metagenomic approach. It used just classical NAT techniques to detect and characterized HPgV-1 from serum samples. Authors based their research on previous metagenomic data. Thus, title should be modified. On the other hand, authors do not cite the referred article on metagenomic in patients with hepatic failure from which HPgV-1 reads were detected. If not published, include a brief description of the objective and the main findings.
Authors mentioned that serum samples were pooled to be tested. How? How many individual samples were put together per each pool?. Obviously, as authors detected, this would generate a dilution effect, underestimating the real frequency. Discuss this.
Important: A real time PCR calibration curve of HPgV-1 is required. Authors point out an interesting observation in lines 164-175 that needs to be supported by experimental data. In addition, appropriate citations should be included in this paragraph.
“The nested- PCR is not adequate for viral diagnosis due to the detection of very low viral loads” . What did you mean?
Remove Figure 1A. Include a table that summarizes real-time and nested-PCR results, Ct values for each sample.
GenBank accession number for each sequence is required (include it on the phylogenetic tree, also).
Amplicon size is missing both in the draft and in the figures 1 and 2. In the figure 2, samples from Brazil needs to be highlighted and bootstrap values lower 70% should be hidden.
Include a supplementary table with the information of the sequences retrieved from GenBank.
Lines 135 and 159: That is not prevalence. Frequency, instead.
Figure 2. Substitution model is not indicated.
Lines 198-200: Extensive local genetic diversification is the rule for RNA viruses and do not necessarily imply a co-evolution with the host. Many information gaps are still needed to be uncovered. Be cautious and consider all plausible hypotheses.
Lines 200-204: Please rephrase. What you believe in is an observation. This genotype is predominant in Brazil.
Moderate editing needed
Author Response
REPLY TO THE REVIEWERS COMMENTS/CRITICISMS
REVIEWER #3
In this manuscript, authors present a study aimed to detected and characterize HPgV-1 form 27 patients diagnosed with severe hepatic failure. This work continues a supposed previous work in which, by a metagenomic approach, several reads belonging to HPgV-1 were detected.
We are very grateful to reviewer#3 for the positive feedback regarding our work.
HPgV-1 has been studied as a likely commensal virus, not specifically associated to a particular medical condition or pathological process, since it has been also identified in healthy individuals.
Further research is needed to clarify the potential (pathogenic and/or benefic) role of this agent in certain conditions. Several reports focused on HPgV-1 detection and characterization in several populations and health conditions has been published from Brazil. Even though this work might be of interest it needs to be substantially improved before considering it for publication.
Major comments
This work was not performed with a metagenomic approach. It used just classical NAT techniques to detect and characterized HPgV-1 from serum samples. Authors based their research on previous metagenomic data. Thus, title should be modified. On the other hand, authors do not cite the referred article on metagenomic in patients with hepatic failure from which HPgV-1 reads were detected. If not published, include a brief description of the objective and the main findings.
We thank reviewer #3 for the valuable recommendations. The title was properly modified in order to avoid causing misunderstanding, since metagenomics is not the focus of this manuscript. Thus, the metagenomics study is yet to be published, and it comprehends the application of this methodology in samples of hepatic failure patients whose common infections (Hepatitis B, C, etc) were not detected by conventional methods. Thus, viral metagenomics was performed through NGS to find potential viral causes and we found some interesting results, including HPgV-1 (which was directly confirmed in this study) and other important pathogenic agents but this was not an objective of this study. We accordingly performed changes in the manuscript title.
Lines 2-3, Page 1 “ New title : “... Human Pegivirus-1 (HPgV-1) detection and genotyping in Brazilian patients with fulminant hepatitis…”
We also gave a brief introduction of the previous study that was the base for the HPgV-1 detection studies “... During a previous metagenomic study identifying viral causes in cases of FH with no known etiology, we found universal presence of HPgV-1 sequence reads. This finding raised questions related to the prevalence and circulating genotypes among those pa-tients. Therefore, a total of 27 plasma samples were tested for HPgV-1 RNA using RT-qPCR, sequencing and phylogenetic analysis.…”
Lines 209-210. We deleted this phrase from the text “... In conclusion, we observed HPgV-1 reads in samples obtained from patients with FH. …”
Authors mentioned that serum samples were pooled to be tested. How? How many individual samples were put together per each pool?. Obviously, as authors detected, this would generate a dilution effect, underestimating the real frequency. Discuss this.
For the metagenomic study, we collected 27 plasma samples from patients with diagnosis suggestive of fulminant hepatitis but who were negative for the commonly tested hepatotropic viruses. From all collected samples and in order to reduce the cost of the sequencing we assembled four pools, one of which contained 6 samples and the rest contained 7 samples. That said, in order not to lose genetic material from the pools, we concentrated and extracted the whole pool volume using a sensitive kit for extraction of large volumes (High Pure Viral Nucleic Acids Kit, Roche). Thus, despite the assembly of several samples due to the applied concentration and extraction techniques we prevent the dilution effect of the samples. Additionally, the results were confirmed through RT-PCR testing directly in the samples, not the pools which additionally prevents negative results and helps to identify the index samples that represented the objective of this study.
Important: A real time PCR calibration curve of HPgV-1 is required. Authors point out an interesting observation in lines 164-175 that needs to be supported by experimental data. In addition, appropriate citations should be included in this paragraph.
We appreciate the suggestion of reviewer #3 on the requirement of an inclusion of RT-PCR calibration curve. This can help to estimate the sensitivity of the test. However, the primers and the hydrolytic probe were obtained from an already published manuscript and are largely used for the real-time detection of HPgV-1 RNA. Additionally, the goal of this study was only to detect and genotype the HPgV-1 present in the index patients with diagnosis of fulminant hepatitis. However, the suggestion of the reviewer is helpful for further studies that aim to quantify the viral load of HPgV-1 in a wide range of clinical samples that can help to elucidate molecular aspects of this infection.
“The nested- PCR is not adequate for viral diagnosis due to the detection of very low viral loads” . What did you mean?
We are grateful for the valuable comment of reviewer#3.
By using this phrase we meant that nested-PCR can detect very low quantities of circulating viral nucleic acids without clinical significance. We removed this phrase from the revised version of the manuscript:
Lines 177-180. The following phrase was removed from the revised version of this manuscript “... . The nested-PCR is not adequate for viral diagnosis due to the detection of very low viral loads. That is the reason why real-time PCR detected 4 positive samples and by nested-PCR we detected an additional of 8 samples that were HPgV-1 positive.
…”
Remove Figure 1A. Include a table that summarizes real-time and nested-PCR results, Ct values for each sample.
We are grateful for the valuable comment of reviewer#3.
We included a table incorporating the results obtained by real-time PCR and nested PCR for each positive samples. We left the figure because it reflects the results obtained by the diagnostic procedures. The table is presented below
|
Table 01. RT-PCR and conventional PCR results from HPgV-1 positive samples |
|||
|
Sample ID |
Positivity for the specific method |
||
|
Conventional nested-PCR |
RT-PCR |
Cycle threshold |
|
|
HF01 |
+ |
_ |
NA* |
|
HF04 |
+ |
+ |
23.6 |
|
HF05 |
+ |
+ |
23.1 |
|
HF06 |
+ |
_ |
NA* |
|
HF07 |
+ |
_ |
NA* |
|
HF09 |
+ |
+ |
24.3 |
|
HF16 |
+ |
_ |
NA* |
|
HF17 |
+ |
_ |
NA* |
|
HF19 |
+ |
+ |
31.5 |
|
HF22 |
+ |
_ |
NA* |
|
HF24 |
+ |
_ |
NA* |
|
HF26 |
+ |
_ |
NA* |
|
NA: non-applicable; *: did not amplify under these conditions. |
|||
GenBank accession number for each sequence is required (include it on the phylogenetic tree, also).
We are grateful for the valuable comment of reviewer 3. We provided the GenBank reference numbers of the obtained sequences at the end of the manuscript:
Lines 233-234, Page 7: “... The obtained 5’UTR HPgV-1 sequences were deposited under the following accession numbers: SRX20084088-SRX20084099. …”
Amplicon size is missing both in the draft and in the figures 1 and 2. In the figure 2, samples from Brazil needs to be highlighted and bootstrap values lower 70% should be hidden.
We are grateful for the valuable comment of reviewer#3.
The amplicon size of the positive samples was 378 bp. This was included in the manuscript:
Line 129, Page 3: “... Nested-PCR amplification of the HPgV-1 amplicon of 378 pb of several samples is shown in Figure 1B. …”
We also excluded the bootstrap values that were below 70%. The new tree was represented in figure 2.
Include a supplementary table with the information of the sequences retrieved from GenBank.
We are grateful for the valuable comment of reviewer#3.
We included a supplementary table with the used reference sequences used for the reconstruction of the HPgV-1 phylogenetic tree. Observe supplement 1.
Line 110, Page 3, “... see Supplementary table 1…”
Lines 135 and 159: That is not prevalence. Frequency, instead.
The term “prevalence” was replaced by “frequency” where appropriate. We are grateful for this suggestion.
Figure 2. Substitution model is not indicated.
The substitution model for this study according to the Bayesian Information Criteria was TVMe+R4. This information was introduced in the modified version of this manuscript.
Lines 131-133, Page 4: “... The best model that was chosen by the Bayesian Information Criteria (BIC) for phylo-genetic reconstruction was the TVMe+R4. …”
Lines 198-200: Extensive local genetic diversification is the rule for RNA viruses and do not necessarily imply a co-evolution with the host. Many information gaps are still needed to be uncovered. Be cautious and consider all plausible hypotheses.
Such a clustering might reflect regional HPgV-1 circulation, since RNA viruses are supposed to present significant diversification but a genotype-specific geographic distribution is usually observed [31]. As an ancient virus, we can raise several hypothesis about its evolutionary history, e.g. co-evolution with the host, but there is yet many information to be studied.
We agree that one of the main factors for the diversification of the RNA viruses is local genetic evolution instead of co-evolution with the host. In that line the previously cited reference of Pavesi et al., 2001 was removed from the revised version of the manuscript. We added this information in the discussion section as we believe that there is still much to be done in order to support our statements in lines 198-200. Therefore, we rephrased the phrase in order to better express the discussion of our findings related to HPgV-1 evolutionary history and our findings.
Lines 197-206, Page 6: “... Studies performed in Brazil, show also that in the Latin American country genotype 2 is a dominant one [9,32,35] including the state of São Paulo [35]. This might be the reason, for the extensive identification of HPgV-1 genotype 2 among the tested pa-tients. An interesting observation in the phylogenetic tree was that the strains obtained from FH patients clustered together with reference Brazilian sequences. Such a clus-tering might be due to a regional HPgV-1 circulation, since RNA viruses present local genetic diversification related genotype-specific geographic distribution [32,36]. Therefore, the genotypic distribution observed in our study might be a reflection of the circulation of the most pronounced genotype in state and in Brazil that is important for studies connected to viral molecular epidemiology.…”
Lines 200-204: Please rephrase. What you believe in is an observation. This genotype is predominant in Brazil.
We performed the suggested modifications as responded in the inquiry above.
- Conclusions
We slightly modified our conclusion section.
Lines 208-209, Page 6, We modified the initial phrase of the conclusion section “... We observed high frequency of HPgV-1 RNA among FH patients; however, we used a combination of strategies that could have detected low circulating viral loads. ..”
Round 2
Reviewer 1 Report
This study by da Silva et al. reports the detection of HPgV-1 in 12 out of 27 Brazilian fulminant hepatitis patients. While the authors have addressed some of the concerns raised previously, there are still several unresolved issues that need clarification before considering the publication of this study.
Comments
1) It appears from the authors' response that the metagenomic analysis was merely an ‘inspiring’ preliminary analysis that provided some clues to the presence of HPgV-1 in some clinical samples (Line 155-157), and the primary methods used to obtain meaningful and interpretable results were PCR and sanger sequencing. Thus, I see no compelling reasons for including this analysis in the manuscript. Moreover, since the authors did not cite any work in the sentences mentioning the metagenomic analysis, I can only guess that the work is also unpublished or has not gone through a peer-review process. Besides this, the current version of the Materials and Methods section lacks any description of the metagenomic sequencing and analysis, suggesting that the authors themselves also do not consider it an integral part of this study. Based on the currently presented results, removing all information related to the metagenomic analysis should actually improve the readability of the work, and make it less distracting. Simply stating that the virus was detected by RT- and nested-PCR should be sufficient without losing any meaningful results.
2) I previously raised a concern that the reported mean Ct value (28.5 ± 7.3, Line 28) was untraceable as the Ct values were reported only for some positive samples. However, the authors have now clarified that Ct values are not available for the 8 samples tested positive for the virus solely through nested-PCR amplification (Table 1). Therefore, a straightforward calculation would suggest that the mean Ct value should be (23.6+23.1+24.3+31.5)/4 = 25.625. I’m unsure how the mean value of 28.5 ± 7.3 was derived. Please explain.
3) Figure 1B is still poorly labelled. Please label each lane appropriately, and provide clear indications of where the 378-bp amplicon should be located on the gel.
4) The authors state that Figure 1B displays “positive HPgV-1 amplicons of 378 bp” (Line 130). However, it is clear that the bands in lanes 4, 6, and 8 do not at all match with the band in the “C+” lane (which I can also assume is a positive control). Please discuss why the results should be considered positive despite the band mismatches.
5) In addition, Figure 1B shows multiple bands in each lane. Please discuss and address the presence of these multiple bands.
6) The authors reported that 4 samples were identified as positive for HPgV-1 by RT-PCR (Line 124-125 and Figure 1A), and with the nested-PCR approach, a total of 12 samples were detected positive for the virus (Line 127-129). However, it appears that there are only a maximum of 7 samples presented in Figure 1B. These numbers do not add up. Please explain.
7) The phylogeny (Figure 2) is still poorly labelled with illegible branch support values, and tip names. The scale bar label is also too small. If the authors intend to include this information on the tree, it should be displayed clearly; otherwise, they should explore some other approaches to present the information on the tree. In addition, since the authors discuss the geographical locations of the sequences on the tree (Line 137-140, and 201-202), the authors should present them on the tree as well. Please add. Lastly, the authors have now clarified that they rooted the tree using the mid-rooting method. Please provide a justification for choosing this method. If the method cannot be justified, consider changing the tree's rooting method and provide a rationale. Alternatively, an unrooted tree could be used instead.
8) The statement “All samples that were characterized as subgenotype 2B formed a separate cluster and were basally located compared to sequences obtained from Brazil, the USA and Egypt” (Line 136-138) is incorrect. The tree presented in Figure 2 clearly shows that the authors' sequences cannot be considered basal. Please correct.
9) Please justify the use of the term “prevalence”, which is present throughout the manuscript, from the study design / methodological point of view.
10) The reference for the statement “[5´UTR] is well known to be able to differentiate HPgV-1 genotypes from 1 to 7.” (Line 83-84) is still missing.
11) The statement that “[genotype 2] is the most prevalent in the world” (Line 136) is also missing a reference for support.
12) The NCBI SRA database and FASTQ format are not suitable for Sanger sequences (Line 234-235). Please change.
13) I still find the writing difficult to follow in many places due to occasional grammatical and punctuation errors. There are also still many instances where words are chosen poorly. The authors may seek a professional language editor to assist their writing.
I still find the writing difficult to follow in many places due to occasional grammatical and punctuation errors. There are also still many instances where words are chosen poorly. The authors may seek a professional language editor to assist their writing.
Author Response
REPLY TO THE REVIEWERS COMMENTS/CRITICISMS
REVIEWER #1
This study by da Silva et al. reports the detection of HPgV-1 in 12 out of 27 Brazilian fulminant hepatitis patients. While the authors have addressed some of the concerns raised previously, there are still several unresolved issues that need clarification before considering the publication of this study.
We are grateful for the Reviewer #1 feedback. Response to the raised concerns is given below.
COMMENTS
1) It appears from the authors' response that the metagenomic analysis was merely an ‘inspiring’ preliminary analysis that provided some clues to the presence of HPgV-1 in some clinical samples (Line 155-157), and the primary methods used to obtain meaningful and interpretable results were PCR and sanger sequencing. Thus, I see no compelling reasons for including this analysis in the manuscript. Moreover, since the authors did not cite any work in the sentences mentioning the metagenomic analysis, I can only guess that the work is also unpublished or has not gone through a peer-review process. Besides this, the current version of the Materials and Methods section lacks any description of the metagenomic sequencing and analysis, suggesting that the authors themselves also do not consider it an integral part of this study. Based on the currently presented results, removing all information related to the metagenomic analysis should actually improve the readability of the work, and make it less distracting. Simply stating that the virus was detected by RT- and nested-PCR should be sufficient without losing any meaningful results.
We are grateful for the valuable comment of Reviewer#1.
We understand the reviewer's concerns, however, we point out the basement that served to perform the presented study independent of the fact that these data were still not published.
In accordance with the reviewer suggestion, we removed the following paragraph from the “Materials and Methods” Section:
Lines 122-126 “... The performed study was based on previous metagenomic identification of HPgV-1 reads by next-generation sequencing in pooled plasma samples belonging to patients with FH (range of reads between 1198 to 8725). This led to investigations into quiries on the molecular detection of HPgV-1 and its genotypes in such patients, among other molecular aspects associated withto the HPgV-1 infection. …”
2) I previously raised a concern that the reported mean Ct value (28.5 ± 7.3, Line 28) was untraceable as the Ct values were reported only for some positive samples. However, the authors have now clarified that Ct values are not available for the 8 samples tested positive for the virus solely through nested-PCR amplification (Table 1). Therefore, a straightforward calculation would suggest that the mean Ct value should be (23.6+23.1+24.3+31.5)/4 = 25.625. I’m unsure how the mean value of 28.5 ± 7.3 was derived. Please explain.
We are grateful for the reviewer correction regarding the mean value. It seems that the error originated from our intention to calculate the median value. We now present in the text the corrected mean value of the cycle threshold obtained in this study that was 25,625 (±3,9).
Line 124, Page 3: “… (mean 25.6±3.9) …”
3) Figure 1B is still poorly labelled. Please label each lane appropriately, and provide clear indications of where the 378-bp amplicon should be located on the gel.
We are grateful for the valuable comment of Reviewer#1.
In the revised the old version of the figure and we identified all samples. The negative samples were also identified including the case where non-specific amplification was observed. The modified figure is presented now in the new version of the manuscript (Line 140, Page 4).
4) The authors state that Figure 1B displays “positive HPgV-1 amplicons of 378 bp” (Line 130). However, it is clear that the bands in lanes 4, 6, and 8 do not at all match with the band in the “C+” lane (which I can also assume is a positive control). Please discuss why the results should be considered positive despite the band mismatches.
We are grateful to the reviewer for the valuable comment.
We will respond to the reviewer by the following way: depending on the viral load, the intensity of band fluorescence can be different, as samples with higher viral load can appear slightly lower and brighter compared to the positive control (in this particular case probably shows lower viral concentration). Additionally, oscillations in the electric current and the buffers may cause these differences. We emphasize however, that this is not a s significant difference. Additionally, all positive samples were sequenced and confirmed by phylogeny that they belong to HPgV-1 which undoubtedly reveals that these are positive cases.
5) In addition, Figure 1B shows multiple bands in each lane. Please discuss and address the presence of these multiple bands.
These bands represent non-specific amplification. Once the positive samples were confirmed by sequencing, there is no need for further investigation of the non specific amplification. .
6) The authors reported that 4 samples were identified as positive for HPgV-1 by RT-PCR (Line 124-125 and Figure 1A), and with the nested-PCR approach, a total of 12 samples were detected positive for the virus (Line 127-129). However, it appears that there are only a maximum of 7 samples presented in Figure 1B. These numbers do not add up. Please explain.
We presented only one agarose gel with some of the positive samples in this manuscript and in the figure. As we performed several gel electrophoresis, it is impossible and unnecessary to present all of the gels in this manuscript.
7) The phylogeny (Figure 2) is still poorly labelled with illegible branch support values, and tip names. The scale bar label is also too small. If the authors intend to include this information on the tree, it should be displayed clearly; otherwise, they should explore some other approaches to present the information on the tree. In addition, since the authors discuss the geographical locations of the sequences on the tree (Line 137-140, and 201-202), the authors should present them on the tree as well. Please add. Lastly, the authors have now clarified that they rooted the tree using the mid-rooting method. Please provide a justification for choosing this method. If the method cannot be justified, consider changing the tree's rooting method and provide a rationale. Alternatively, an unrooted tree could be used instead.
We are grateful for the valuable comment of the reviewer.
We totally modified our tree, making it more visibile. In order to show the path of evolution of HPgV-1, we root our tree because it gives an indication of the direction or evolutionary change across time (Kinene et al., 2016). Our tree was rooted using the Midpoint method, which calculates tip to tip distances and then places the root between the two longest tips (Swofford et al., 1996). This method is often applied in viral datasets when we intend to search for an outgroup; in our scenario, we were able to cluster the viral sequences into genotypes by Midpoint Rooting. We also showed the geographic location in the newly reconstructed phylogenetic tree by using differential coloring.
8) The statement “All samples that were characterized as subgenotype 2B formed a separate cluster and were basally located compared to sequences obtained from Brazil, the USA and Egypt” (Line 136-138) is incorrect. The tree presented in Figure 2 clearly shows that the authors' sequences cannot be considered basal. Please correct.
We are grateful for the valuable comment of Reviewer #1.
We removed the term basally located from the discussion section. Now the discussion reads like this:
Lines 136-137, Page 4 “... 2B formed a separate cluster with sequences obtained from Brazil, the USA, and Egypt …”
9) Please justify the use of the term “prevalence”, which is present throughout the manuscript, from the study design / methodological point of view.
We believe that the correct term is rather frequency. We changed throughout the manuscript “prevalence” to “frequency”.
10) The reference for the statement “[5´UTR] is well known to be able to differentiate HPgV-1 genotypes from 1 to 7.” (Line 83-84) is still missing.
We added the reference for the primers that are used to differentiate the HPgV-1 genotypes 1 to 7 (Miao et al.,2017) (Line 86, Page 2)
11) The statement that “[genotype 2] is the most prevalent in the world” (Line 136) is also missing a reference for support.
We searched for references that describe the prevalence of genotype 2. It is in fact more prevalent in Europe (Castro M et al., 2022), and the Americas (Silva ASN et al., 2022). This was stated in the text:
Lines 197-198, Page 6: “… Genotype 2 is a prevalent genotype of HPgV-1, being the most disseminated in Europe and the USA [30-32]. …”
12) The NCBI SRA database and FASTQ format are not suitable for Sanger sequences (Line 234-235). Please change.
We apologize that we previously submitted the sequences to a short read database. Now the sequences were submitted to the GenBank. Their GenBank accession numbers are the following: OR227612- OR227623. This was added at the end of the manuscript (Line 239, Page 7).
13) I still find the writing difficult to follow in many places due to occasional grammatical and punctuation errors. There are also still many instances where words are chosen poorly. The authors may seek a professional language editor to assist their writing.
The text was edited by a native speaker. The certificate is attached in the new submission.
REFERENCES
Castro M, Matas IM, Silva E, Barradas PF, Amorim I, Gomes H, Monteiro Á, Nascimento MSJ, Mesquita JR. Occurrence and molecular characterization of human pegivirus-1 (HPgV-1) viremia in healthy volunteer blood donors from Northern Portugal. J Med Virol. 2022 Jul;94(7):3442-3447. doi: 10.1002/jmv.27687. Epub 2022 Mar 8. PMID: 35229315.
Kinene T, Wainaina J, Maina S, Boykin LM. Rooting Trees, Methods for. In: Encyclopedia of Evolutionary Biology. Elsevier; 2016:489-493. Accessed June 9, 2023. http://dx.doi.org/10.1016/b978-0-12-800049-6.00215-8
Miao, Z.; Gao, L.; Song, Y.; Yang, M.; Zhang, M.; Lou, J.; Zhao, Y.; Wang, X.; Feng, Y.; Dong, X.; Xia, X. Prevalence and Clinical Impact of Human Pegivirus-1 Infection in HIV-1-Infected Individuals in Yunnan, China. Viruses 2017, 9, 28. https://doi.org/10.3390/v9020028.
Silva ASN, Silva CP, Barata RR, da Silva PVR, Monteiro PDJ, Lamarão L, Burbano RMR, Nunes MRT, de Lima PDL. Human pegivirus (HPgV, GBV-C) RNA in volunteer blood donors from a public hemotherapy service in Northern Brazil. Virol J. 2020 Oct 14;17(1):153. doi: 10.1186/s12985-020-01427-6. PMID: 33054824; PMCID: PMC7556973.
Swofford DL, Olsen GJ, Waddell PJ, Hillis DM. Phylogenetic inference. In: Hiillis DM, Moritz D, Mable BK, editors. Molecular Systematics. Sinauer Associates; Sunderland, MA: 1996. pp. 407–514.
Reviewer 3 Report
All major issues were responded/considered.
Additional minor points
.Delete line 173.
. Genbank accesion numbers need to be included in the figure caption also.
Minor editing requiered
Author Response
REPLY TO THE REVIEWER COMMENTS/CRITICISMS
REVIEWER #3
All major issues were responded/considered.
We are grateful that the reviewer approved the presented response letter and the performed changes. Thank you!
Additional minor points
We appreciate Reviewer #3 feedback and the opportunity to address the additional minor points you raised, thank you.
Delete line 173.
Regarding your suggestion to delete line 173, we decided to modify the phrase like that.
Lines 175-176, Page 6: “... A limitation of our study was that we did not include an accompanying negative control. …”
Genbank accesion numbers need to be included in the figure caption also.
We are sorry for missing this information and have properly included the GenBank accession numbers in the Figure 2 caption to improve the traceability of our results. (Lines 154-155, Page 5).